# A Self-Assembled Aggregate Composed of a Fatty Acid Membrane and the Building Blocks of Biological Polymers Provides a First Step in the Emergence of Protocells

**DOI:** 10.3390/life6030033

**Published:** 2016-08-11

**Authors:** Roy A. Black, Matthew C. Blosser

**Affiliations:** 1Department of Bioengineering, University of Washington, Seattle, WA 98195, USA; 2Department of Chemistry, University of Oxford, Oxford OX1 3TA, UK; matthewblosser@gmail.com

**Keywords:** origin of life, prebiotic, self-assembly, amphiphiles, fatty acid, vesicle, nucleoside, peptide, oligonucleotide, membrane

## Abstract

We propose that the first step in the origin of cellular life on Earth was the self-assembly of fatty acids with the building blocks of RNA and protein, resulting in a stable aggregate. This scheme provides explanations for the selection and concentration of the prebiotic components of cells; the stabilization and growth of early membranes; the catalysis of biopolymer synthesis; and the co-localization of membranes, RNA and protein. In this article, we review the evidence and rationale for the formation of the proposed aggregate: (i) the well-established phenomenon of self-assembly of fatty acids to form vesicles; (ii) our published evidence that nucleobases and sugars bind to and stabilize such vesicles; and (iii) the reasons why amino acids likely do so as well. We then explain how the conformational constraints and altered chemical environment due to binding of the components to the membrane could facilitate the formation of nucleosides, oligonucleotides and peptides. We conclude by discussing how the resulting oligomers, even if short and random, could have increased vesicle stability and growth more than their building blocks did, and how competition among these vesicles could have led to longer polymers with complex functions.

## 1. Introduction

How did prebiotic molecules on the early Earth assemble into machinery (proteins) and stores of information (RNA) surrounded by a membrane? How each of these structures assembled and how they became co-localized remain unclear. All three structures are composed of simple building blocks that can be generated by abiotic processes [1]. Therefore any explanation for the origin of cells requires solving two problems: (1) how were the building blocks selected and concentrated as required for the formation of the three structures; and (2) how did the membranes, RNA, and protein become associated with each other?

We have proposed that interactions among the building blocks of the three structures, prior to the formation of RNA or proteins, can answer both of these questions [2]. The heart of our proposal is that the building blocks self-assembled into an aggregate with RNA and protein components bound to a fatty acid membrane, and that this aggregate stabilized the membrane and facilitated the formation of the two polymers. In more detail, we hypothesize:
**a**.Fatty acids self-assembled in water to form a membrane.**b**.Components of RNA and protein were selected and concentrated via binding to self-assembled fatty acid membranes.**c**.These bound building blocks stabilized fatty acid membranes against salt-induced flocculation and increased the rate of vesicle formation.**d**.Membranes that were more stable bound more building blocks, leading to an auto-amplifying system.**e**.The resulting aggregate facilitated the formation of nucleosides, oligonucleotides and peptides, both because of the selection and concentration of building blocks and because of the conformational constraints and altered chemical environment due to binding.**f**.The oligomers, initially composed of random sequences, stabilized membranes and induced membrane growth more effectively than their unjoined components did, leading to the accumulation of oligonucleotides and peptides prior to the evolution of their complex functions in metabolism and information transfer.

This model (Figure 1, filled arrows) offers a simple explanation for the selection and concentration of the prebiotic components of cells; the stabilization and growth of early membranes; the formation of the two biopolymers; and—unlike any other scheme we are aware of—the occurrence of membranes, RNA and protein in a single unit. Indeed, we view this co-localization as a clue to the origin of cells, rather than as the result of a random event that followed the formation of these three structures. In our scenario, membranes, protein and RNA emerge together because the prebiotic building blocks for all three self-assemble into a stable aggregate.

This scheme relates to other perspectives on prebiotic chemistry in several ways. It is consistent with the suggestion that RNA was not the first informational polymer but rather evolved from an earlier structure with different components. This suggestion stems from the observation that the sugar, bases and inorganic linker of RNA are remarkably well-suited to the structural requirements of a self-coding polymer [3,4]. Our scheme would apply equally well to building blocks of any evolutionary predecessor of RNA, as long as these building blocks, too, co-aggregate with fatty acids.

The scenario is broader in some respects than other proposals that the membrane was the starting point or played an active role in the emergence of cells [5,6,7]. Objections to these proposals have included the lack of a compelling explanation for how RNA and proteins became associated with the membranes, although some progress in this regard has recently been made. Szostak’s group broke new ground by showing that certain mineral surfaces stimulate membrane formation as well as increase the polymerization of nucleotides [8,9], but this work did not address how nucleotides arose or how proteins were incorporated into protocells. Ruiz-Mirazo [10] and Sutherland [11] have recently advanced the case for a systems chemistry approach to the origin of protocells to replace the more typical approach of explaining one structure at a time. However, neither of these papers offer a mechanism to explain the association of amphiphiles with bases, sugars and amino acids. Deamer and colleagues moved the field forward by showing that a phospholipid matrix can organize and polymerize mononucleotides [12,13] (with caveats discussed below), and they have described how functional systems could emerge in hydrothermal pools from hydration-dehydration cycling of vesicles encapsulating polymers [14].

A key feature of our scenario, unlike the previous membrane-centered proposals, is the focus on how the components of all three essential protocell structures were brought together. Affinities among the components in aqueous solution would complement their concentration in evaporating pools. In addition, differential affinities provide a plausible explanation for the selection of monomers found in biopolymers, and our scheme highlights the catalytic potential of a fatty acid membrane associated with RNA and protein building blocks. We propose that such a membrane could catalyze nucleoside synthesis and facilitate the condensation reactions required for the formation of both peptides and oligonucleotides; for example, nucleoside formation in this scheme would have resulted from the joining of ribose and bases bound to a vesicle surface, rather than from reactions between compounds smaller than bases and sugars as proposed by Sutherland. By identifying an integral role for fatty acid membranes in the selection, concentration and oligomerization of the building blocks of RNA and proteins, we strengthen the argument that the pathway to cells began with the self-assembly of amphiphiles.

The rest of this review is organized as follows. In Section 2, we describe the characteristics of the building blocks of RNA and protein that would enable them to spontaneously bind to fatty acid membranes. For each component, we then review the current evidence that such binding does occur (scenario elements **a** and **b**) and that membranes that bind the building blocks are more stable than membranes composed solely of amphiphiles (scenario elements **c** and **d**). We also discuss the potential for additive or synergistic interactions among building blocks. In Section 3, we discuss how the binding of RNA and protein building blocks to fatty acid membranes could help explain both the synthesis of nucleosides and the formation of short oligomers of nucleotides and of amino acids (scenario element **e**). This section is more speculative than the previous one, and we suggest how to pursue these ideas experimentally. In Section 4, we discuss, still more speculatively, how our scheme could also explain the accumulation of longer oligonucleotides and peptides prior to evolution of their complex biological functions (scenario element **f**). At this point, the scenario merges with later stages in protocell evolution suggested by Szostak (e.g., [15]) and Damer and Deamer [14].

As is the case with most proposed prebiotic chemistry scenarios, this scheme may inadequately account for the complexity of the early Earth environment. The impossibility of reproducing the full range of compounds and conditions means that some reactions that proceed in laboratory settings would not proceed in early-Earth conditions, and vice versa. Nonetheless, the more aspects of the scheme that we validate experimentally, the stronger the case will be that it describes one possible route to protocells. Other routes, including those that do not involve lipids at an early stage, are reviewed in [5,6].

## 2. Formation and Stabilization of a Self-Assembled Aggregate of Prebiotic Building Blocks

### 2.1. Amphiphiles

Amphiphiles are molecules with a hydrophilic “head” group and a hydrophobic “tail”. In water, amphiphiles can spontaneously aggregate into small clusters called micelles or larger structures called vesicles (Figure 2). Vesicles are cell-like compartments, composed of an aqueous core separated from the environment by a bilayer of amphiphiles. Multilamellar vesicles have one or more additional bilayers inside.

The first membranes were probably composed of simple amphiphiles such as fatty acids, which consist of a hydrocarbon chain (the “tail”) of various lengths with a terminal carboxyl group (the “head” group) (Figure 2) [16,17]. Fatty acids as short as eight carbons spontaneously form vesicles in water at neutral pH [18]. These molecules are made by abiotic reactions likely to have occurred on the ancient Earth [17,19], and are found in meteorites [20]. Fatty acid vesicles grow by incorporating free or micellar fatty acids from the surrounding solution [8,21,22]. Thus prebiotic fatty acids could have self-assembled into cell-like compartments with the capacity for growth.

One concern regarding an early role for fatty acids in protocell membranes has been that the short-chain species that probably predominated [23] form vesicles only at relatively high concentrations. This concern is reduced by the recent finding that small percentages of long-chain fatty acid species substantially increase vesicle formation [24]. Moreover, concentrations either at surfaces or in evaporating pools would have been much higher than in the prebiotic ocean as a whole [14]. Experimentally, decanoic acid is generally used as a prototypical prebiotic fatty acid because it is long enough to form vesicles readily [25] and short enough to have been found in high abundance [20].

Two additional problems have weakened the case for an early role of fatty acid vesicles in the emergence of protocells: First, they flocculate in the presence of low concentrations of divalent ions or moderate (~0.3 M) concentrations of NaCl [26]. Long-chain alcohols and glycerol monoesters stabilize vesicles under these conditions [18,27,28], but the prebiotic availability of these compounds is uncertain. A fresh water origin for protocells has been proposed as another solution [14]. Second, as noted in Section 1, other proposals emphasizing a primary role for fatty acids have lacked a compelling explanation for how RNA and proteins became associated with prebiotic membranes. Our proposed scenario, in which the *building blocks* of RNA and protein spontaneously bound to and stabilized fatty acid aggregates and *subsequently* oligomerized, solves both of these problems. The remainder of Section 2 presents the rationale and evidence for the association of RNA and protein building blocks with fatty acid membranes. In both this section and Section 3, the focus is on associations with the surfaces of fatty acid aggregates, whether unilamellar vesicles, micelles or multilamellar structures as indicated, rather than on entrapment of solutes in the aqueous space inside vesicles.

### 2.2. Bases

RNA is composed of a ribose-phosphate backbone with one of four nitrogenous heterocyclic bases attached to each ribose (Figure 3). In this section we present the rationale and evidence for the binding of bases to fatty acid surfaces, and in Section 2.3 we address the binding of ribose.

The nucleobases (the bases found in RNA and DNA) have been identified in meteorites [29] and/or are generated by plausibly prebiotic reactions [1]. Their structural features suggest how these bases might interact with a lipid bilayer. They share structural characteristics of planarity and of both hydrophobicity and hydrophilicity. Planarity could enable the insertion of bases into a lipid bilayer without excessive disruption of the packing of the hydrocarbon tails. The simultaneous hydrophobicity and hydrophilicity of the bases [30] would enable them to interact with both the headgroups and the hydrocarbon core of a fatty acid membrane.

To test whether nucleobases do in fact interact with fatty acid membranes, we employed three independent assays [2]. Our most compelling result was that nucleobases are retained with decanoic acid micelles in a filtration-based assay. In this assay, unbound compounds pass through a filter that retains the micelles and micelle-bound components. The nucleobases were generally retained to a greater extent than related bases that are not incorporated into RNA (Figure 4). In a second assay, we found that adenine dialyzes more slowly in the presence of decanoic acid vesicles than in the presence of acetic acid, which is too short to form vesicles. In the third assay, we found that including adenine in the aqueous solution below a monolayer of stearic acid in a Langmuir trough altered the surface pressure of the monolayer, even though adenine alone is not surface active.

Thus nucleobases bind to fatty acid membranes as proposed in element **b** of our scenario. Do they also stabilize vesicles as proposed in element **c**? As noted above, NaCl at 0.3 M flocculates decanoic acid vesicles [2,26]. We found that adenine virtually eliminates this flocculation (at ≥32 °C) [2]. Other bases also reduce flocculation, and the magnitudes of the inhibitory effects correlate with the extent to which the bases bind decanoic acid aggregates in the filtration assay [2]. Elements **b** and **c** of our scenario therefore hold true for nucleobases.

Another possible interplay between fatty acid vesicles and bases, which we have not yet explored, is formation of Watson-Crick pairs within the membrane or between the lamellae of multilamellar vesicles. Neither bases nor nucleosides (e.g., adenosine and uridine) pair in aqueous solution [3,30], but base pairing between monomers has been detected in nonaqueous media [31]. Interactions with a fatty acid bilayer (or bilayers in the case of a multilamellar vesicle) could confine and align the bases, thereby allowing base pairs to form. If such base pairing stabilizes vesicles, this phenomenon would help explain how bases that are capable of pairing were selected by protocells.

### 2.3. Sugars

Ribose and related sugars were probably available prebiotically. These compounds are generated by the formose reaction, which starts with simple prebiotic compounds [32]. The products tend to react with each other rather than remaining in solution and ribose is not typically a major product. However, interaction with borate preserves the products, particularly ribose [32,33]. Alternately, ribose may have been delivered to the prebiotic Earth via interstellar ice grains. A recent study demonstrated that sugars including ribose form when analogs of interstellar ice are irradiated, apparently through a formose-type reaction [34].

A distinguishing feature of sugars is an abundance of hydroxyl groups. The aldehyde of a linear aldose and one of the hydroxyl groups can react to generate a ring with a new hydroxyl group, at the “anomeric” carbon, that increases the conformational complexity since it can point up or down (in Haworth projections) (Figure 5). We speculate that the hydroxyl groups of a sugar can hydrogen-bond with the carboxyl head groups of a fatty acid vesicle, and if so, it is likely that some configurations of hydroxyl groups hydrogen-bond more strongly than others. These interactions could stabilize the vesicle, as proposed in element **c** of our scheme. Consistent with this idea, glycerol esterified with a fatty acid stabilizes fatty acid vesicles, possibly due to interactions between the two free hydroxyls of the glycerol and the carboxyl headgroups of the fatty acids [27,28,35].

To determine whether sugars do stabilize fatty acid vesicles, we tested whether they inhibit salt-induced flocculation as described in Section 2.2. We found that a number of sugars do inhibit flocculation of decanoic acid vesicles. Ribose is more effective than glucose and even xylose, a diastereomer of ribose (Figure 5). The difference in efficacy between closely related sugars suggests that the inhibition of flocculation is due to binding rather than merely to an effect on the bulk properties of the solution; the latter would not be expected to vary with minor differences in the sugars’ structures.

### 2.4. Amino Acids and Dipeptides

Ten amino acids (glycine, alanine, valine, leucine, isoleucine, proline, serine, threonine, aspartate and glutamate) and even some dipeptides (Figure 6) are generally considered to have been prebiotically available [36,37]. The designation of prebiotic is based on various lines of evidence including identification in meteorites and synthesis under plausibly prebiotic conditions such as the Miller-Urey sparking experiments [1,38,39].

Amino acids and dipeptides could interact with a fatty acid membrane in at least five ways: (1) amine groups could hydrogen bond with fatty acid carboxyl groups; (2) if protonated, amines could bond electrostatically to the carboxyl groups; (3) the carboxyl group of the amino acid or dipeptide could hydrogen bond with a fatty acid carboxyl group; (4) the hydrophobic sidechain (R_1_ and R_2_ in Figure 6) of an amino acid such as leucine could interact with the hydrocarbon core of the membrane; and (5) the hydrophilic sidechain of an amino acid like serine or threonine could interact with the fatty acid carboxyl groups.

Several studies suggest the plausibility of such interactions. Decylamine, composed of an amine group attached to a chain of ten carbons, stabilizes decanoic acid vesicles at both high and low pH, suggesting that the amine, whether or not protonated, interacts with the fatty acid carboxyl groups [40]. A hydroxyl group attached to a chain of ten carbons dramatically stabilizes decanoic acid vesicles [18]. As noted in Section 2.3, the free hydroxyl groups of a glycerol monoester increase vesicle stability. Finally, lysine (albeit not a prebiotic amino acid) associates with palmitic acid to form a foam [41].

To our knowledge, there are no published studies of whether single amino acids or unmodified, prebiotic dipeptides interact with fatty acid bilayers. Two studies did show interaction between fatty acid vesicles and dipeptides that included a nonprebiotic amino acid (phenylalanine or *O*-methyltyrosine) and that had the ends blocked by acetylation of the amine and amidation of the carboxyl group [42,43], and the interaction of nonprebiotic amphipathic peptides with oleic acid vesicles has been reported [44]. Other studies have demonstrated interactions between phospholipid membranes (which are not considered prebiotic) and hydrophobic peptides, again containing a nonprebiotic amino acid and blocked ends [45,46]. Separately, Pohorille has modeled longer peptides interacting with phospholipid membranes [47].

These studies, along with the analysis of possible modes of interaction and the studies with compounds related to amino acids, support our proposal that prebiotic amino acids and dipeptides can interact with a fatty acid bilayer. In addition to such direct interactions, association with nucleobases or sugars bound to the membrane would increase the number of ways in which amino acids and dipeptides could bind to a vesicle. To investigate whether unmodified prebiotic amino acids and dipeptides do in fact bind to and stabilize fatty acid aggregates, we propose employing the same methods used to demonstrate interactions with nucleobases and sugars.

### 2.5. Simultaneous Binding of Multiple RNA and Protein Building Blocks to Fatty Acid Membranes

Just as bound bases and sugars could increase amino acid and dipeptide binding, the inverse is also possible, and indeed a key feature of our scenario is that more than one type of molecular building block binds to a fatty acid membrane at a time. In support of this proposition, we found that the inhibition of salt-induced flocculation of decanoic acid when both adenine and ribose are present is greater than the inhibitory effects of either alone, and consistent with the sum of their independent inhibitory effects. This finding suggests that the two compounds do bind to fatty acid vesicles simultaneously [2].

## 3. Formation of Oligomers

Section 2 addressed how the building blocks of fatty acid membranes, RNA, and protein might have become co-localized in a stable aggregate. In this section, we discuss how this co-localization could have facilitated the formation of the two biological polymers (element **e** of our scheme).

Several challenges arise in explaining the abiotic formation of RNA and protein. Association of their building blocks with a fatty acid membrane helps overcome all of these challenges. (i) How building blocks were concentrated sufficiently to react with each other is plausibly solved by their binding to a membrane. Even low energy binding of reactants to a soft interface is sufficient to increase reaction rates, as demonstrated in emulsified water droplets [48]; the authors of this study note that “the process is based on an interfacial reaction-diffusion mechanism that is expected to be quite general”; (ii) Indiscriminate incorporation of all possible prebiotic bases, sugars and amino acids into polymers would have impeded the formation of reproducible, functional sequences. The selective binding of building blocks to a fatty acid membrane would eliminate this problem; (iii) Even with sufficient concentrations of a limited number of reactants, polymerization would have required catalysts to bring the reactants together and to constrain their conformations in ways that favored the formation of specific covalent bonds. Both the hydrophobic core and hydrophilic surface of an amphiphilic bilayer plausibly fulfill these functions. In addition, membranes could incorporate components such as dipeptides (as discussed in Section 2.4) which could have contributed to the catalytic activity of the bilayer. In support of this idea, plausibly prebiotic dipeptides catalyze at least two reactions important for protocells, the stereospecific synthesis of sugars [49] and the aminoacylation of an RNA [50]; (iv) Finally, the formation of both phosphodiester and peptide bonds requires the generation of a water molecule, and the reactions are not thermodynamically favored in an aqueous environment. Amphiphilic bilayers and multilamellar structures provide an environment with reduced water activity, which may lessen this barrier [51].

Most of these potential roles of membranes have been noted by others in various contexts [52]. Several experiments, which will be discussed in Section 3.1 and Section 3.2, have indeed found that membranes can increase the formation of oligonucleotides from mononucleotides and of peptides from activated amino acids. However, in these experiments neither the monomers nor, in most cases, the amphiphiles employed were plausibly prebiotic. Below we describe in more detail the status of efforts to produce the two biological polymers by plausibly prebiotic mechanisms, and the critical role a bilayer composed of prebiotic amphiphiles and prebiotic building blocks of RNA and protein could have played.

### 3.1. RNA

The first step in explaining the origin of RNA is understanding the formation of nucleosides (Figure 7), and much work has been devoted to possible abiotic mechanisms. Since the bases and ribose are confirmed or likely prebiotic compounds, as discussed in Section 2, the simplest explanation for the formation of nucleosides is that a reaction occurred between preformed bases and preformed ribose. However, the requisite glycosidic bond forms inefficiently under all prebiotic conditions tested over the past 40 years [53,54,55]. Recent efforts have succeeded in synthesizing pyrimidine [56] and purine [57] nucleosides by starting with compounds smaller than bases and sugars, although the prebiotic plausibility of the sequences of reactions employed has been questioned [58]. Nonbiological nucleosides are easier to synthesize [59,60,61,62], and they offer a further solution if there is an evolutionary path from them to the biological molecules.

Our scheme focusses on the possibility of formation of the glycosidic bond by a reaction between a membrane-bound base and membrane-bound ribose. This possibility is of particular interest because it would help to explain the selection of the bases and sugar in RNA and the co-localization of RNA with membranes. As reviewed by Sutherland in detail [54], there are several reasons why formation of the glycosidic bond is difficult. Taking adenine and ribose as example reactants, the tautomers of the two compounds that favor the reaction are relatively minor. Moreover, while protonation of the anomeric hydroxyl of ribose makes it a better leaving group, the acidic conditions required for its protonation make the adenine *less* nucleophilic. We suggest that the observed binding of adenine and ribose to fatty acid vesicles [2] could solve these problems by inducing favorable conformations; the reactants could bind either to apposing surfaces in a multilamellar vesicle or to the same membrane surface. Interaction of the adenine and/or ribose with other components such as amino acids and dipeptides could also affect the frequency of the required tautomers. To test for glycosidic bond formation in the presence of fatty acid vesicles, we propose incubating a mixture of adenine, ribose and decanoic acid over a range of temperatures, pHs and salt concentrations, with or without dehydration.

Explaining the prebiotic polymerization of nucleosides to form RNA is as challenging as explaining the generation of nucleosides [3,4,55,58,63]. After nucleosides are formed, the next step is the addition of phosphate as a linker; phosphate may have been derived prebiotically from the mineral schreibersite [64]. In order to polymerize with regiospecificity, the resulting nucleotides must be oriented properly [63]. Certain mineral surfaces apparently fulfill the need for orientation, but virtually all studies to date in this area have employed artificially activated (i.e., not prebiotically plausible) nucleotides [65]. A more promising approach employs repeated drying and rehydration of a solution of nucleotide monophosphates with phospholipids [12,62] or salts [66]. Strikingly, drying adenosine monophosphate in the presence of a multilamellar phospholipid matrix organizes the monomers between lamellae in a way that appears to facilitate polymerization [67]; a caveat to this finding is that the combination of low pH and high temperature employed can lead to loss of some bases from the phosphate-ribose backbone [68].

We hypothesize that the orientation of nucleotides required for polymerization could also occur on the surface of a fatty acid vesicle or between the lamellae of a multilamellar fatty acid vesicle, due to interactions of hydroxyl groups on the sugar with the headgroups of fatty acids. Such interactions and/or the presence of plausible prebiotic catalysts such as amino acids and dipeptides may eliminate the need for low pH. We also suggest that once a polymer formed in this way, it would be positioned to serve as a template for a complementary strand. This suggestion is plausible because when nucleosides covalently attached to fatty acids form a monolayer at an air-water interface, they are capable of base pairing with nucleosides in the aqueous phase [69,70]. Thus aggregation with fatty acids provides sufficient structural scaffolding to align free nucleosides. Whether free nucleosides and/or nucleotides bind directly to fatty acid aggregates could be investigated by the methods used to demonstrate binding by nucleobases. To test for the polymerization of mononucleotides in our proposed system, we suggest using methods similar to those employed in the studies with phospholipids [12,68].

### 3.2. Proteins

Proteins are chains of amino acids joined by peptide bonds. These bonds result from a condensation reaction between amine and carboxyl groups (Figure 6). Peptide bonds form under putative prebiotic conditions ranging from dry heat to cold eutectic solutions [38,71,72,73,74]. An alternative pathway toward proteins involves depsipeptides, which form more easily than peptides [75].

Our proposal that binding of amino acids to a bilayer facilitates peptide bond formation is supported by work with nonprebiotic systems. Several studies have shown that phospholipid membranes increase peptide bond formation between nonprebiotic amino acids [45,46,76]. A conceptually related study demonstrated peptide bond formation at an air-water interface, with an esterified form of the amino acid [77]. More relevant to our proposal, Adamala and Szostak showed that an oleic acid membrane increases the Ser-His-catalyzed formation of a peptide from amidated leucine and esterified acetyl-phenylalanine. They speculate that the reactants may “partition to the membrane, which allows the reaction to occur at the solvent-lipid bilayer interface, or even within the bilayer, and thereby minimize ester hydrolysis”. They also note that “membrane localization of [the amino acid substrates] may … decrease the pKa of the N-terminal amino group, and so enhance its reactivity by increasing the fraction of nucleophilic deprotonated amine” [42]. In another study with a fatty acid system, Murillo-Sanchez et al. present evidence for acid-base catalysis of peptide formation between amidated leucine and an activated tyrosine-like compound [43]. Finally, Furuuchi et al. reported that decanoic acid vesicles increase synthesis of di- and triglycine from glycine in high-pressure chambers at temperatures above 100 °C [78]. 

These studies, together with the general consideration of the catalytic potential of membranes found at the beginning of Section 3, support our hypothesis that prebiotic lipid vesicles can facilitate peptide bond formation between unmodified prebiotic amino acids. To test this hypothesis, we propose incubating such amino acids with decanoic acid over a range of temperatures, pHs and salt concentrations, with or without dehydration.

## 4. From an Assemblage of the Components to an Evolving Protocell

How did assemblies composed of a membrane, nucleobases, sugars, amino acids, and short oligomers become evolving protocells? A major step in the pathway to protocells would have been the emergence of polymers sufficiently long to carry out such functions as replication, acquisition of additional building blocks, and cell division. In the final element of our scheme, we propose that this step could have been accomplished as follows:
Some short oligomers catalyzed condensation (i.e., covalent linkage) between building blocks to form new oligomers, including at least some copies of itself.The presence of oligomers increased the stability, growth or division of vesicles to a greater extent than monomers.Vesicles bearing catalytic oligomers would have a higher overall concentration of oligomers, and would accumulate fatty acid and biopolymer building blocks at the expense of vesicles that lack catalytic oligomers.If the condensation mechanism joined short oligomers as well as single building blocks, longer oligomers would accumulate (assuming the reaction was faster than hydrolysis).

Several lines of experimental evidence support the first three points above, and point D follows logically. With respect to point A, the demonstrated ability of dipeptides to catalyze organic reactions is discussed in Section 3. In support of point B, Szostak and colleagues have shown the following: (i) Oligonucleotides are retained within fatty acid vesicles and can osmotically stress them such that they grow at the expense of unstressed vesicles [15]; (ii) An acetylated, amidated hydrophobic dipeptide (acetyl-phenylalanine-leucine-amide) that inserts into the bilayer can increase vesicle growth at the expense of vesicles lacking this peptide [42]; (iii) Oxidized di-cysteine induces pearling of filaments in multilamellar vesicles, giving rise to daughter vesicles [79,80]; (iv) An enzyme that catalyzes formation of a certain fatty acid derivative inside fatty acid vesicles can thereby stabilize the vesicles against Mg^2+^ [81]. Regarding point C, the Ser-His dipeptide catalyzes formation of the dipeptide acetyl-phenylalanine-leucine-amide, just mentioned, that increases vesicle growth [42]. The extent to which a dipeptide that catalyzed condensation between two amino acids would have to prefer synthesis of itself in order to propagate is beyond the scope of this review; a sophisticated treatment of the analogous problem for a self-replicating RNA has been published by Higgs and colleagues [82].

Once polymers accumulated that were sufficiently long to carry out complex functions such as replication, acquisition of additional building blocks, and cell division, then Darwinian evolution could begin. As others have argued [37], the first polymers with evolved complex functions must have been RNAs or molecular predecessors of RNA that could propagate by template-directed replication, and the coding of protein sequences by RNA was presumably a later development. A distinct implication of our scheme, stemming from its emphasis on the potential of even random peptides to increase vesicle stability and growth, is the following: RNAs that facilitated the production even of noncoded peptides would have conferred a selectable advantage on protocells. One way an RNA could increase peptide production is to activate amino acids by acylation. The subsequent evolution of RNAs that aligned their activated amino acid with other, similarly activated amino acids by base-pairing at a distal site with a “guide” RNA could have led to coded protein synthesis. As described by Damer and Deamer [14], the emergence of such activating and template RNAs in a small number of fatty acid vesicles (or possibly only one) is conceivable given the vast number of vesicles that can form in even a small volume of water.

## 5. Conclusions

The concept of a self-assembled fatty acid membrane binding the building blocks of RNA and protein helps answer two fundamental questions regarding the origin of protocells: how the components of the polymers were selected and concentrated, and why the polymers emerged together in a membranous package. Experimental evidence shows that nucleobases and sugars bind to and stabilize fatty acid membranes, and that there are sound reasons to predict that amino acids and dipeptides do so as well. These associations of the building blocks with a fatty acid bilayer could have overcome barriers to abiotic formation of oligonucleotides and peptides, which in turn led to more competitive vesicles and ultimately to the package of two polymers and a membrane that took over the world.

## Figures and Tables

**Figure 1 life-06-00033-f001:**
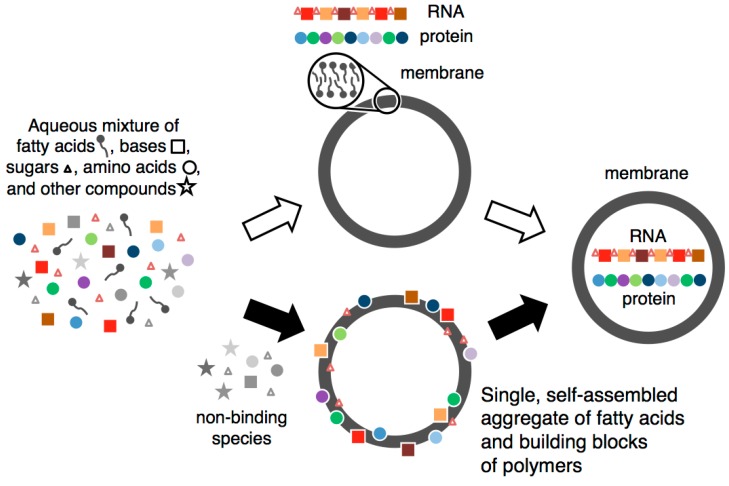
Proposed scheme for origin of protocells. Conventional (open arrows) and proposed (filled arrows) paths to a protocell.

**Figure 2 life-06-00033-f002:**
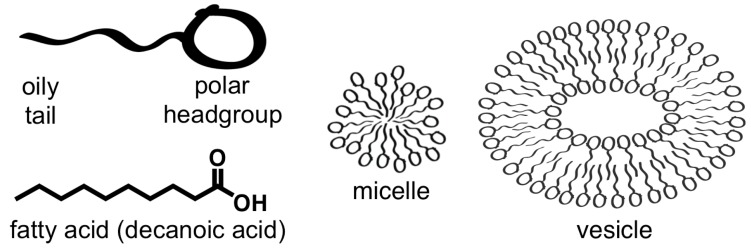
Self-assembly of a simple amphiphile. Fatty acids self-assemble into micelles and into bilayer structures called vesicles that resemble cell membranes. Provided by Sarah L. Keller.

**Figure 3 life-06-00033-f003:**
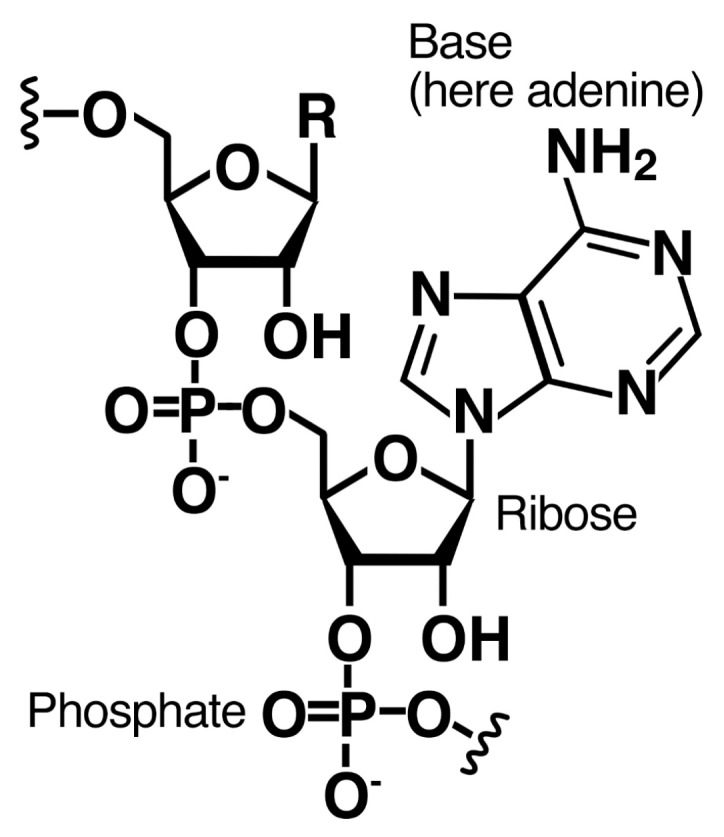
Structure of RNA. The organic building blocks of RNA are the sugar (ribose) and four bases (adenine, guanine, uracil and cytosine).

**Figure 4 life-06-00033-f004:**
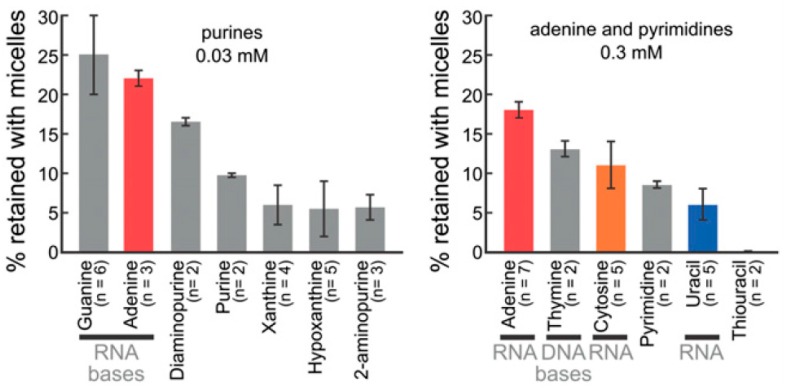
Decanoic acid micelles selectively bind heterocyclic nitrogenous bases. The solution to which bases were added contained 180 mM decanoic acid, pH 8.25. Reproduced from Reference [2]. Scatchard analysis of additional data for adenine suggests two modes of binding, one with a *K*_d_ of about 11 µM and one with much lower affinity [2].

**Figure 5 life-06-00033-f005:**
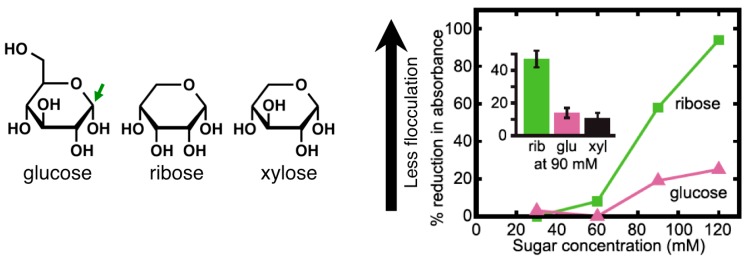
Interaction between sugars and fatty acid vesicles. The green arrow indicates the anomeric carbon. The pyranose rather than furanose forms and α rather than β anomers are shown to emphasize the potential for hydrogen bonding with a surface. The addition of ribose to a solution of decanoic acid vesicles inhibits salt-induced flocculation of vesicles more effectively than addition of glucose or xylose. The sugars were added to solutions containing 80 mM decanoic acid/100 mM bicine/pH 7.9; NaCl was added to 300 mM. Graph reproduced from Reference [2]; structures provided by Sarah L. Keller.

**Figure 6 life-06-00033-f006:**
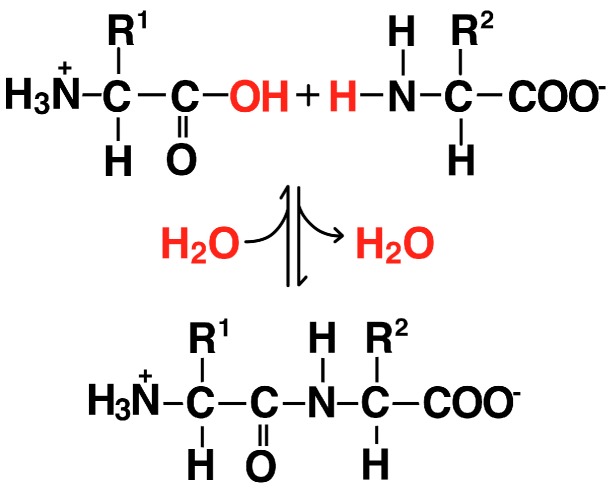
Two amino acids (**top**) join via a peptide bond (**bottom**). Proteins are long, linear polymers of amino acids.

**Figure 7 life-06-00033-f007:**
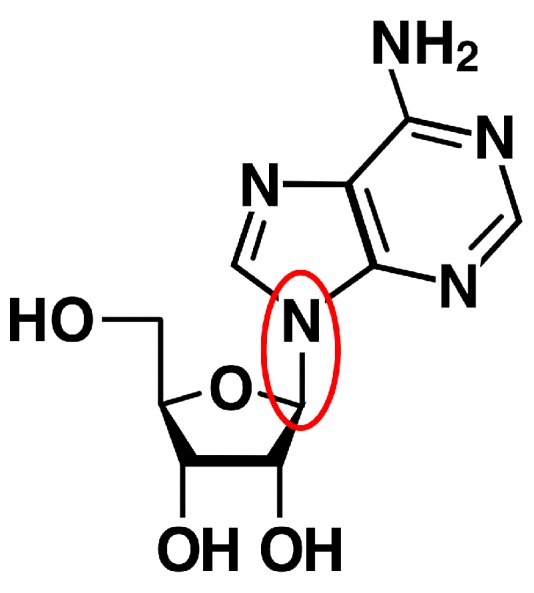
One of four units from which RNA is built. Adenosine is the nucleoside composed of adenine and ribose. The glycosidic bond is circled.

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
