# Peer review of "A Self-Assembled Aggregate Composed of a Fatty Acid Membrane and the Building Blocks of Biological Polymers Provides a First Step in the Emergence of Protocells"

_life, 2016, doi:10.3390/life6030033_

Round 1
Reviewer 1 Report
This paper presents a hypothesis that aggregates of fatty acids, nucleic acids and proteins all together were important for the origin of life. There are a lot of interesting ideas here that seem quite plausible, however the paper goes a long way based on rather little experimental evidence. I don't think that this should prevent publication of the paper, but I do think that the authors should be more careful to distinguish which parts of the argument have experimental support and which do not. It would also be beneficial to be up-front and say something like "this part of the argument needs further investigation and it could be studied in the following way."
The authors show that decanoic acid flocculates at high salt concentration. They therefore argue that the nucleotides and peptides are important to prevent flocculation. But this does not necessarily follow. One could argue that life needed to begin at low salt where the decanoic acid is stable, or that life needed to use some other form of lipid that was more stable than decanoic acid, or that life did not need lipids at all to begin with.
In Fig 4, I am not sure how to interpret the % retained with micelles. How strong binding is 25%? The solution of nucleotides is flowing through a filter. So it is not in equilibrium. If there were some equilibrium partition between the micelles and a solution it might be possible to estimate the strength of binding in terms of free energies.
In Fig 5, the drawing of ribose is a six-atom ring instead of the more usual 5-atom ring with the 5' carbon above the ring. Is there a reason for this?
There is some discussion of the binding of bases to fatty acids and also of sugars to fatty acids, but not of nucelosides and nucleotides. This raises lots of questions. If nucleosides/tides were formed separately, would they bind better or worse than the bases and the sugars? Also, the phosphorylation state of the nucleotides is probably important - especially the charges on the phosphates are likely to influence association with the charged lipids.
The paper seems to hint that the fatty acids bind both ribose and bases separately and bring them together. Does this make the formation of nucleosides more likely? Are you arguing that fatty acids catalyse the formation of nucleosides? Has this been shown experimentally?
The paper is not too clear what kind of structure is envisaged for the lipids - micelles (with molecules bound to the surface), vesicles (with trapped molecules inside), or lamellar phases (with molecules sandwiched between). Please clarify this.
Reviewer 2 Report
In this work, Black and Blosser expand on their recent PNAS paper showing stabilization of fatty acid assemblies by nucleobases, ribose, and (newly presented here) amino acids and peptides. This research program is at a fairly early stage, however, the early results are very interesting. The authors have presented their work here in a hypothesis paper, which is a perfect venue for this interesting, not yet fully characterized, model. I recommend publication with a few corrections, additions of references, and suggestions, listed below.
The article generally strongly favors the model of nucleoside formation wherein they are formed by condensation from preformed nucleobases and ribose, vs. the Sutherland/Powner/Szostak pathway in which nucleotides are formed in a single reaction. The authors acknowledge the competing theory, but only fairly late in the paper. I am personally undecided/agnostic about which is correct, but I think that the paper would read more clearly (and draw in more skeptical readers) if Sutherland's model was mentioned early on, and, as the authors note, that their results provide support for that originally explored by Orgel, and, more recently, Hud.
Re: potential base pairing in the membrane/between lamellae: Williams, Chawla, and Shaw (Biopolymers Volume 26, Issue 4, pages 591–603, April 1987) is a good reference for base pairing in non-aqueous solution.
Pg 7 line 224, typo, "prebioticly"
Pg 7 lines 228-230, another explanation that is invoked for this is reduced surface charge (likely particularly important for high-salt environments).
Pg 7 line 242, I would present the sugars as either the beta anomers (thermodynamically favored in solution in each case) or with a squiggly line to denote both are present. Certainly not incorrect the way it's presented, though.
I would add a line to the figure caption that pyranosyl forms are shown in each case here (again, not incorrect, but readers are likely used to seeing the beta-furanosyl form of ribose in particular, since that anomer is trapped in nucleosides).
Page 8 line 252, "indicates" -> "suggests." I agree this is likely the case but I think it can't be stated with certitude unless there is a direct line of evidence for this.
Page 8 line 260, typo, "prebioticly"
Page 9 line 318, another relevant paper from the Szostak group about this: http://onlinelibrary.wiley.com/doi/10.1002/ange.201505742/full
(Could also be mentioned earlier where peptide/membrane localization is discussed)
Page 11 line 378 typo, "prebioticly"
Where the Deamer lipid-assisted polymerization model is mentioned throughout, I would add one or two references to the recent work from Rajamani (https://www.mdpi.com/2075-1729/5/1/65) showing some of the shortcomings of this model system - namely, abasic site formation. This, similarly, will be true of any system that requires acidic conditions, particularly for adenosine, which is the canonical nucleoside most prone to abasic site formation.
Page 12 line 406, the peptide used in this work was Ser-His, not His-Ser.
Page 12 line 426, proofreading mark still present
Page 12 line 442, the osmotic stress induced growth observed in this study was found even with 5'-UMP, and shorter RNAs (1-40mers).
Reviewer 3 Report
Excellent combination of a review and new data that results in a novel and compelling approach to the origins of cellular life. I wish I had written it myself!
Author Response
We thank the reviewer for his comment.